# Unravelling the Mechanisms of Manipulating Numbers in Language Models

## Abstract

Recent work has shown that different large language models (LLMs) converge to similar and accurate input embedding representations for numbers. These findings conflict with the documented propensity of LLMs to produce erroneous outputs when dealing with numeric information. In this work, we aim to explain this conflict by exploring how language models *manipulate* numbers and quantify the lower bounds of accuracy of these mechanisms. We find that despite surfacing errors, different language models learn interchangeable representations of numbers that are systematic, highly accurate and universal across their hidden states and the types of input contexts. This allows us to create universal probes for each LLM and to trace information — including the causes of output errors — to specific layers. Our results lay a fundamental understanding of how pre-trained LLMs manipulate numbers and outline the practical potential of more accurate probing techniques in addressed refinements of LLMs' architectures.

## 1 Introduction

One major limitation of large language models (LLMs) is their inability to guarantee factually precise and correct outputs. This is a fundamental trait due to the probabilistic nature of transformer models, and thus not a solvable problem, as has recently been emphasized by (Kalai et al., 2025). However, knowing the extent of the problem (and whether it can be mitigated) is a feasible task; one that is important given the recent ubiquity of LLMs in research and broader society.

Mathematical problems provide an ideal playground in which to assess the current ability of LLMs to manipulate precise information. Because it operates through a set of formal rules and produces computationally verifiable outputs, mathematics provides the opportunity to rate not only the tendency of a model to output incoherent information (Sun et al., 2024), but also to examine in detail the coherence of the reasoning steps outputted by a given LLM. While this research direction is rife with challenges, ranging from manipulating implicitly visual data in geometry problems (Ahn et al., 2024) to mapping plain language onto specific types of problems (Duchnowski et al., 2025), it has also been met by genuine successes, e.g., as an aid to solve mathematical problems (Li et al., 2025).

A large body of work has highlighted limitations of LLMs when it comes to arithmetic tasks (Xu et al., 2025; Kvinge et al., 2025; Satpute et al., 2024; Lee et al., 2025; Bertolazzi et al., 2025; Nikankin et al., 2025), as well as proposed means to alleviate these limitations centered on re-designing numeric representations to be principled and precise (e.g., Charton, 2022; Feng et al., 2024; Golkar et al., 2023). More recent developments have suggested that the issues models encounter with arithmetic need not be tied to the quality of their representations of numbers. Several works have attempted to characterize the types of representations models rely on to perform arithmetic computations (Wennberg & Henter, 2024; Zhou et al., 2024; Zhu et al., 2025; Kadlčík et al., 2025). This research direction has proposed that number representations stand out in terms of their structure, with Zhou et al. (2024) and Kadlčík et al. (2025) arguing that models rely on sinusoidal structures, whereas Zhu et al. (2025) suggest that representations are better described using a linear model than a multi-layer perceptron. These results have also been combined with observations as to how models process and succeed at arithmetic tasks.

In this paper, we provide evidence that LLMs converge to systematic and accurate *sinusoidal* representations that are maintained throughout the model in different forms. These representations are largely equivalent across LLMs of different sizes and families. Building upon this, we find that

numeric information from LLMs can be accurately extracted using universal probes and show that more accurate numeric probes allow us to attribute up to 94% of model errors in arithmetical reasoning to particular layers, overriding the existing, correct results.[1]

## 2 RELATED WORK

**Math capabilities of LLMs**  The seminal remarks of Brown et al. (2020) that LLMs that are large enough develop arithmetic capabilities have ushered in a large body of work aimed at evaluating the mathematical capabilities of LLMs (Hendrycks et al., 2021; Cobbe et al., 2021; Sun et al., 2024; Yu et al., 2024) or lack thereof (Nikankin et al., 2025). This in turn has encouraged a focus on how to bolster the capabilities of LLMs or interpret how LLMs perform computations (Zhang et al., 2024; Stolfo et al., 2023). Most relevant to ours, some of the previous work especially focuses on how numbers should be represented in principle. For instance, Charton (2022) assess an impact on models performance in linear algebra problems when employing different encoding schemes based on scientific notation on linear algebra problems. Golkar et al. (2023) propose to encode numeric values by incorporating a scaled, learned control token <NUM>. Feng et al. (2024) remark that the precision of the numeric representation type impacts performance on arithmetic tasks, and argue to use the logarithmic-precision architecture of Feng et al. (2023).

**Numeric representations in LLMs**  The increased focus on evaluating LLM performance in verifiable domains such as math problems has also led to increased interest in the mechanisms by which LLMs compute arithmetic functions. Nanda et al. (2023) demonstrate how a model trained from scratch on modular addition relies on trigonometric operation: it maps inputs onto a unit circle, corresponding to a specific rotation, and then learns to combine the two rotations to derive a valid solution. Kantamneni & Tegmark (2025) apply circuit analysis to highlight how general pretrained models perform addition, and highlight that representations are mapped onto a helix that can be manipulated using trigonometric operations. Zhou et al. (2024) refines these observations in terms of Fourier components (since Fourier transformations map arbitrary functions unto combinations of periodic trigonometric functions), and highlight the distinct role of attention and feedforward sublayers. Focusing more narrowly on representations rather than processing, Zhu et al. (2025) argue that non-linear probes do not provide a better fit than linear probes. Levy & Geva (2025) remark that it is possible to retrieve the digit (value mod 10) of numeric inputs with high accuracy. Kadlčík et al. (2025) propose a linear probe architecture that factors in the sinusoidal nature of the embeddings of numbers, and show that they can retrieve numeric precision with high accuracy.

In summary, there is a growing body of evidence stating that models rely on trigonometric representations for numbers. Here, we build upon Kadlčík et al. (2025) specifically, as the high accuracy of their proposed probe provides new opportunities and novel research directions — as we find, also in tracking and verifying the accuracy of a model's *inner* representations throughout its computations, robust across contexts, model types and operations.

## 3 UNIVERSALITY OF SINUSOIDAL REPRESENTATIONS OF NUMBERS

### 3.1 MODELS LEARN EQUIVALENT REPRESENTATIONS OF NUMBERS

The first point we address is the distinctiveness of LLMs' representations of numbers. Results from Kadlčík et al. (2025) and Zhu et al. (2025) suggest that different models converge to the same *type* of number representations, but they do not provide a direct assessment of how *closely* the representations match across models – which could evidence that a shared representation is a causal consequence of architectural bias and optimization process rather than a coincidental artifact. Following Kadlčík et al., we focus on the input embeddings from eight LLMs of diverse sizes and families with open-sourced, pre-trained checkpoints: OLMo 2 (OLMo et al., 2025), Llama 3 (Grattafiori et al., 2024) and Phi 4 (Abdin et al., 2024).

To quantify whether models converge to similar embeddings, we start by conducting a simple Representational Similarity Analysis (RSA; Kriegeskorte et al., 2008) across the input embeddings for

---

[1]We make all our methods and analyses available to any use at the project repository: `https://github.com/prompteus/numllama`

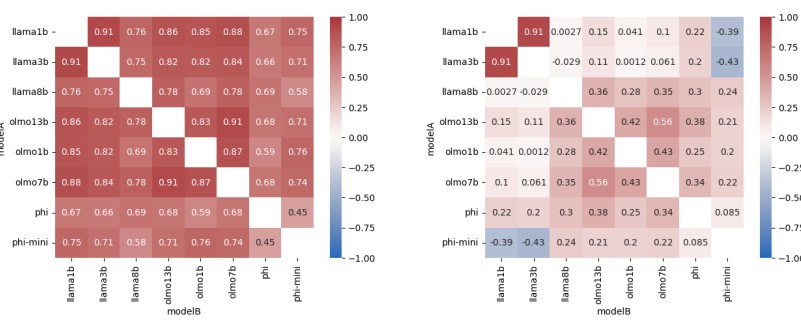

(a) Cosine-based RSA for number tokens     (b) Cosine-based RSA for random tokens

Figure 1: Representational similarity analysis (RSA) scores

number word-pieces in paired models.[2] As a baseline, we compute RSA scores for a random sample of 1000 pieces present in the vocabularies of all the models under consideration. Results of the analysis are displayed in Figure 1. Across all pairs of models, we find that number embeddings systematically yield higher RSA scores than a random sample of word pieces. While some models (especially in the Phi family) are not as well aligned with other models, we do observe high scores both within and across families, demonstrating that models converge to embedding spaces with equivalent similarity structure.

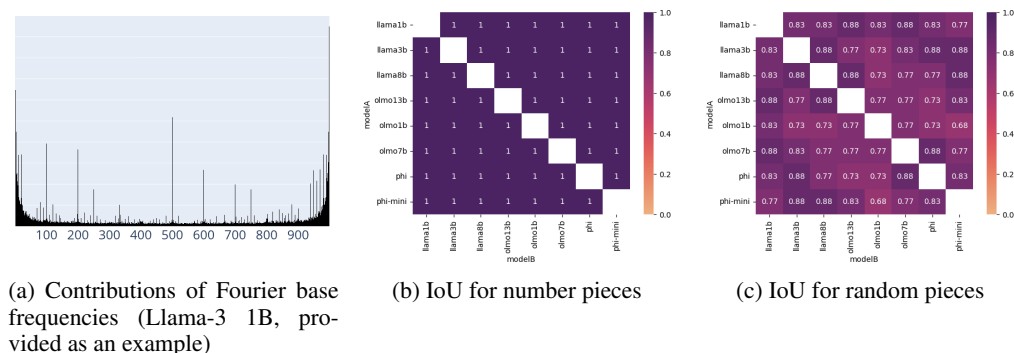

(a) Contributions of Fourier base frequencies (Llama-3 1B, provided as an example)

(b) IoU for number pieces

(c) IoU for random pieces

Figure 2: Intersection-over-union of top $k = 63$ Fourier base frequencies

Another approach to quantifying the similarity of frequential representations characteristic for numbers Kadlčík et al. (2025); Zhou et al. (2024) is by inspecting their base frequencies through Fourier decompositions. Applying a PCA transformation followed by a Fourier transform allows us to quantify the *magnitude* of individual frequencies (Figure 2a). Subsequently, we determine whether two models agree as to how they rank frequencies by computing a simple intersection-over-union (IoU) of the top $k$ frequencies. We repeat this process for every pair of models, using the input embeddings of number pieces (Figure 2b) as well as that of random pieces as previously (Figure 2c). When considering the top $k = 63$ frequencies,[3] we find **perfect agreement across all models** in terms of number pieces.

---

[2]RSA is a second-order similarity measurement. Let $A = \{\mathbf{a}_1, \ldots \mathbf{a}_n\}$ and $B = \{\mathbf{b}_1, \ldots \mathbf{b}_n\}$ be two sets of matching representations, in our case the input embeddings for number word-pieces in two models $\theta_A$ and $\theta_B$. We can assess the extent to which $A$ and $B$ encode the same type of similarity structures by simply computing the (Spearman) correlation $\rho_{\mathbf{s}_A, \mathbf{s}_B}$, where $\mathbf{s}_A = (\cos(\mathbf{a}_1, \mathbf{a}_2) \ldots \cos(\mathbf{a}_{n-1}, \mathbf{a}_n))$ and $\mathbf{s}_B = (\cos(\mathbf{b}_1, \mathbf{b}_2) \ldots \cos(\mathbf{b}_{n-1}, \mathbf{b}_n))$ are two vectors tracking all pairwise similarities within $A$ and $B$ respectively.

[3]See Section A.1 for details how this value of $k$ is found.

In summary, our analyses show that the LLMs from different families learn to represent numbers in a similar topology and trigonometric features using the same dominant frequencies. This is specific to numbers – we do not reproduce these observations with other word pieces shared across models.

## 3.2 Number Representations are Always Sinusoidal

In subsection 3.1, we presented evidence that the input embeddings of numbers across different LLMs are highly consistent. In this section, we aim to find whether such representations are also maintained and employed throughout the models.

For qualitative assessment, we visualize the PCA and its Fourier transform of internal activations of numerical tokens in Section A.3. Besides the apparent wave-like pattern, the representations have a sparse Fourier transform, confirming the sinusoidal character. For quantitative analysis, we look at the model's internal representations through the lens of the sinusoidal probe proposed by Kadlčík et al. (2025). This probe was designed to map input embeddings of LLMs into an integer, thus classifying embeddings into predefined range of numeric values. The sinusoidal probe is defined as:

$$f_{\sin}(\mathbf{x}) = (\mathbf{W}_{\text{out}}\mathbf{S})^T(\mathbf{W}_{\text{in}}\mathbf{x}) \tag{1}$$

$$\mathbf{S}_{ij} = \begin{cases} \sin(ie^j 1000/h) & \text{if } j \equiv 0 \mod 2 \\ \cos(ie^{j+1} 1000/h) & \text{if } j \equiv 1 \mod 2 \end{cases}$$

where $\mathbf{W}_{\text{in}} : h \times d$ and $\mathbf{W}_{\text{out}} : h \times d$ are learned parameters, and $\mathbf{S} : h \times 1000$ injects an inductive bias in the classifier towards sinusoidal representations. Unless otherwise stated, we use $d = 100$, $h$ corresponds to the inner dimensionality of the LLM at hand. We build upon an assumption verified by Kadlčík et al. (2025) stating that sinusoidal probes indeed adapt and employ a sinusoidal representation from inductive bias, if a sinusoidal representation is also present in the input embeddings.

We first assess whether sinusoidal probes are the most suitable, i.e., an accurate choice for decoding the *internal* representations of the model. To contextualize the accuracy in terms of sinusoidal quality of the representation, we also evaluate other types of probes used to decode numbers in previous work Feng et al. (2023); Zhu et al. (2025); Kadlčík et al. (2025) as baselines. we train the sinusoidal probe on hidden representations of each layer of Llama 3.2 1B model when processing addition prompts of the form '$x_1 + x_2$', where $x_1$ and $x_2$ are integers. We specifically evaluate whether the value of $x_2$ can be retrieved from the model's hidden representation on each layer. To assess generalization rather than memorization capacity, we *split* the prompts and corresponding representations into training, validation and test sets containing *distinct* extracted numbers ($x_2$).

Figure 3a shows the accuracy of different probes extracting the value of input numbers from representations of each model's layer, evaluated across six different models. The superior accuracy of sinusoidal probe provides an indicator of both (i) the sinusoidal character, and (ii) accuracy of input number representation across layers.

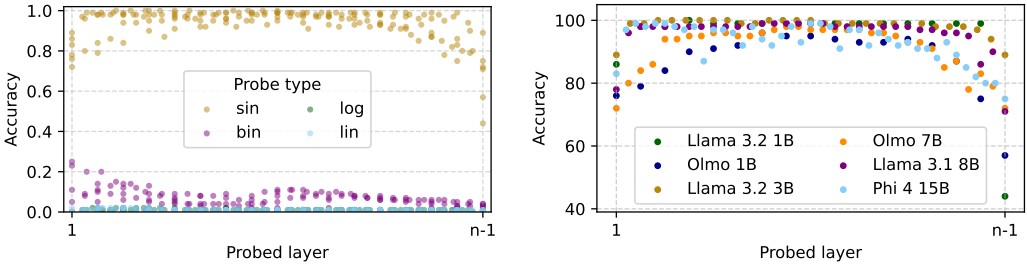

(a) using probes with different bases; math contexts   (b) across different language models; natural contexts

Figure 3: Accuracy of decoding numeric input token from internal activations of language models

Thus far, an important caveat of ours, as well as a methodology of previous work probing representations of numbers in previous work (Section 2) is that we focus on a very specific type of input, namely additions, without any sort of natural language to contextualize them. As such, it is reasonable to assume that the results presented in Figure 3 may not carry to a more natural context: there

is little guarantee that the behavior of a model on arithmetic problems is indicative of its behavior a broad range of applications requiring manipulations of numbers in natural language contexts.

To evaluate the impact of natural language context on accuracy and regularity of numeric representations, we curate a dataset covering four domains involving intensive numeric manipulations: arithmetic (mathematical word problems), temporal (date extraction), medical (ICD-10 diagnostic codes) and culinary (recipe ingredients/quantities). The complete dataset specifications are detailed in Table 1 in Section B. To maximize the quality of learned probes and thus, representativeness of our findings, we also tackle the natural-language class imbalance; as not every number has equal probability to occur in a date or in a recipe, we replace original numbers with integers uniformly sampled from the model vocabulary. We then fit separate probes for activations from every layer, again holding out a set of 100 randomly selected numbers for evaluating generalization.

Figure 3b displays the accuracies of input value extraction from model's hidden states within natural-language contexts across six LLMs of different sizes and three different families. As we can see, the sinusoidal probes reach over 70% of accuracy in all but three cases and over 90% of accuracy for a majority of probing scenarios. Taking into an account that the learned sinusoidal probe also induce a certain error. Therefore, these accuracies present a lower-bound of accuracy of models' representation of input numbers in natural contexts in a sinusoidal representation — showing that the accurate, sinusoidal representation is indeed characteristic and employed by a wide range of recent language models.

A vast majority of cases performing lower than 80% occurs in probing the models' first and a last layer — we analyze these cases in detail later in Section 3.3.

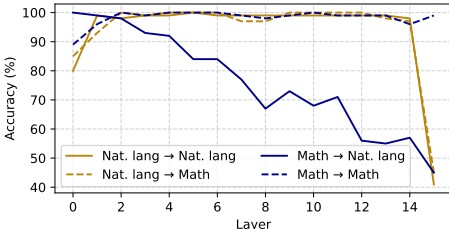

Figure 4: **Generalization of probes** fitted on natural-language occurrences of numeric tokens (solid line), and synthetic, mathematical contexts (dashed line).

A comparison of probes' performance in math an natural-language contexts (Figure 3) may suggest that a type of context does not influence the results of probing accuracy. However, in Figure 4, we show that probes fitted with natural-language contexts are much more *robust* in application across *both* math and natural-language contexts; While both the math and natural-language probes generalize well within their training context type, the natural-language probes are applicable comparably even under a substantial context distribution shift, suggesting that a wider variety of contexts serves as an effective regularization strategy. This observation draws two important implications: First, it informs future work tracing models' internal computations to favor probes trained in natural-language settings. Second, it restrains future work in interpretability from drawing broad conclusions on models' mechanics from fully-synthetic settings, evidencing that such conclusions may not generalize to real-world applications.

In short, our results underscore that the same type of sinusoidal representation of numbers holds in general across different model types. Numbers are represented in a similar, systematic fashion, regardless of which layer or type of contexts we consider.

### 3.3 Different Layers use Interchangeable Numeric Representations

Having found the remarkable degree to which the embeddings for numbers align across different models and the noteworthy universality of sinusoidal representations in varied contexts, we turn to further analyses assessing the extent to which these structures are utilized by the models. First, we address whether the same sinusoidal structure is conserved across a model's computations.

To that end, we employ the probes we developed in Section 3.2 trained separately for each model layer, $L_i$, and measure the accuracy of each probe on representations derived for every *other* layer $L_j \neq L_i$. This approach, while informative, comes with the caveat that probes trained for a given layer $L_i$ might pick up on idiosyncrasies inherent to a specific layer — i.e., the probes might not disentangle what is specific to numbers as opposed to what is specific to a layer. To address this point, we also fit probes using *all but one* layer $(L_1, L_2, \ldots L_{i-1}, \ L_{i+1}, \ldots L_n)$, and evaluate the performance on representations from the held out layer $L_i$. As previously, we hold out a subset of 100 numbers for assessing generalization in validation and test conditions.

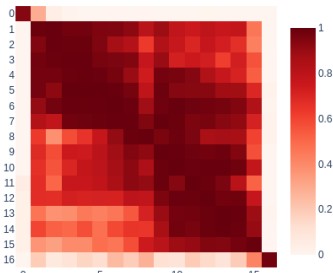

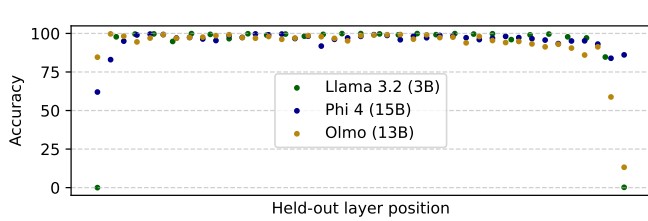

(a) accuracy of probes trained for Llama 1B on a chosen layer (rows) evaluated on all other layers (columns)

(b) probes trained on all-but-one layer, evaluated on the held-out layer

Figure 5: Probes accuracy on activations from unseen *layers*.

Results of probes' cross-layer generalization for Llama 1B are displayed in Figure 5, with largely consistent results for other models in Appendix A.2. We observe that probes fitted on a concrete layer's representations (Figure 5a) generalize outstandingly well to close-by layers, with the exception of the first and the last layer. This trend reveals that language models **operate mostly in a consistent representation** of numbers that is **universal across their computations**, undergoing only minor shifts across layers. This trend is also corroborated by multi-layer probes evaluated in heldout-layer fashion, reaching an accuracy between 95 and 100% across all intermediate layers and three models of diverse sizes and families. This result has practical implications for future work in numeric interpretability — showing that we can train universal, yet highly *accurate* probes for intermediate layers of a broad set of language models[4].

Cross-layer evaluation indicates that models' internal representations tend to differ from the input/output embeddings. We find a notable discrepancy in the *sparsity* of their representation; Whereas the input/output embeddings represent numbers in a more *distributed* fashion, hidden layers use a small number of consistently-ordered sin features.

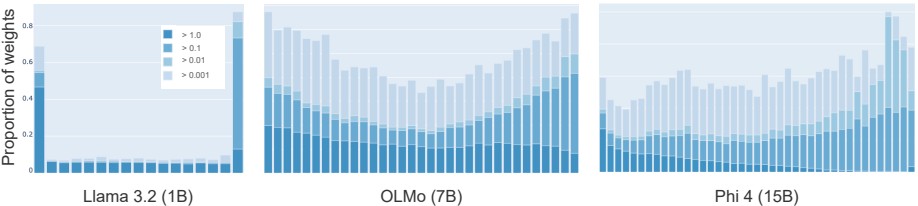

Figure 6: **Difference in probe weights distribution (of $W_{in}$) across layers** shows that the initial and last layers represent numbers in a systematically sparser fashion.

This is visualized in Figure 6, showing that intermediate layers in Llama 1B lead to probes with very few weights with values $> 10^{-5}$.

---

[4]We will release our training scripts, together with reproducibility guidelines for training universal probes, in a final version of this paper

We hypothesize that this discrepancy can be caused by the models' computational and optimization mechanics. As cross-entropy loss is minimized by confident one-hot predictions, models can benefit from using more features to create higher dissimilarity (separability) of tokens. In contrast, internal representations are not subject to direct optimization pressure and might benefit from fewer but more informative features.

Nevertheless, we note that this trend is pertinent across different models to a different extent: OLMo 7B and Phi 15B learn more distributed features also across the intermediate layers — which, in turn, also leads to a better generalization of universal probes (Figure 5b), but still causing an outstanding drop in held-out accuracy compared to other layers. Finally, we note that the models' sparsity profile does not relate to reported performances on arithmetic tasks Kadlčík et al. (2025) and it also does not necessarily determine the accuracy of probes trained specifically for particular layer(s) (Figure 3).

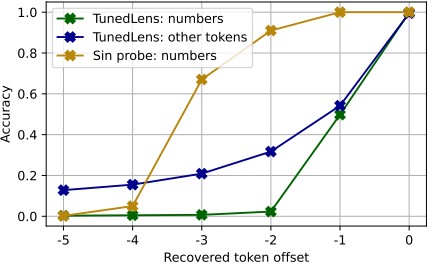 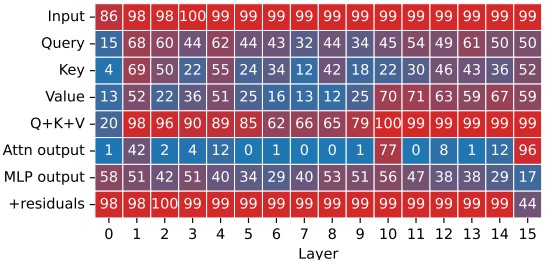

Figure 7: **Accuracy of previous tokens' recovery** for different probes trained on embeddings.

Figure 8: Accuracy of extracting input numbers from each component of the Transformer block using the sinusoidal probe.

Finally, we explore the origin of the high consistency of representations across layers within the internal mechanism that each transformer layer implements. In Figure 8, we visualize the accuracy of probing input representations from each of the components present in transformer layers (Llama 3.2 1B for brevity). We find the accurate numeric representation is scattered across different components in the attention mechanism, with the attention output projection largely violating the sinusoidal representation, which is then, to a large extent, reconstructed in the subsequent fully-connected block. Nevertheless, the consistency of layers' output representation is primarily maintained by residual streams across layers.

To summarize, we find that the same probe can recover the numeric information for representations pooled from different layers of the models we study. Added to our previous observations in Sections 3.1 and 3.2, we can therefore stress that models learn input embeddings for numbers that are **strikingly similar**, regardless of which model they come from. These embeddings are processed into sinusoidal representations that are systematic and consistent regardless of context or hidden layer, with the source of this consistency originating primarily in the residual stream across layers.

## 4 MECHANISMS OF NUMBER MANIPULATION

We have established that models converge to sinusoidal number representations universally. This naturally begs the question of how those sinusoidal representations are used.[5]

### 4.1 MULTI-TOKEN NUMBERS

Our first point of order concerns digits beyond the range of what can be represented in recent LLMs with a single token. To see whether LMs are able to represent even the values of multi-token numbers systematically and accurately, we probe the values of all parts of multi-token numbers from a single

---

[5]In what follows, we focus exclusively on sinusoidal probes for the sake of providing a clearer exposition. Most alternative probe designs suggested by Kadlčík et al. (2025) fail to achieve meaningful performances. For reference, the best performance we observe are from the binary probes in previous-token recovery, achieving 1.2% of accuracy on -1-th recovered token (Figure 7).

representation corresponding to the last numeric token. The accuracy of this experiment uncovers the extent to which the models implement a systematic algorithm of superposing multi-token numbers in a single representation.

For this experiment, we adapt the methodology utilizing natural-language contexts, with a substitution of original numeric values with values spanning two to six numeric tokens, i.e. in a numeric range between $10^3 - 10^{18}$. We use a subset of BBC data that was published after the knowledge cutoff or release date of all the models under consideration (Li et al., 2024).[6]

To be able to contextualize the magnitude of superposition and thus, disentangle a general model mechanism from the mechanism specific for numbers, we also probe representations of natural-language tokens using TunedLens (Belrose et al., 2025). TunedLens is an early-exit interpretability method which aims to explain the contents of intermediate representations by learning to translate them into logits. In practice, for each layer $L_i$, TunedLens involves distilling the computations done in the subsequent layers $L_{i+1}, \ldots L_n$ into a simple affine transformation $\mathbf{W}_i$, such that applying this transformation $\mathbf{W}_i$ to a hidden state at $L_i$ followed by the unembedding matrix closely matches the logits that the LLM would eventually produce. Here, to maximize comparability and align the number and distribution of target categories, we narrow down the space of probed tokens to 1000 tokens (matching the number probes) ranked as 2000–3000thmost-common tokens in our dataset.

Results for Llama 1B and different probing methods are presented in Figure 7. Results of sin probes show that **multi-token numbers are indeed systematically and highly accurately superposed in the latest numeric representation** – reaching an accuracy of 99% for the immediately-preceding number piece (offset -1). However, the accuracy and/or systematicity of the superposition mechanism quickly drops for numbers longer than three tokens (i.e. $\geq 10^9$), reaching close to zero for the 5th preceding token.

The results of natural-language token probing using TunedLens reveal that superpositioning representations of previous tokens is *not* a mechanism specific only to numbers – albeit this mechanism seems more prevalent and accurate for numeric tokens. Aiming to maximise comparability, we also report a comparison of the previous-token recovery using TunedLens for *numeric* tokens — in exactly the same configuration as for the natural-language tokens. This comparison shows that TunedLens is highly inefficient for recovering numeric tokens compared to sinusoidal probes.

## 4.2 ERROR TRACKING

Establishing that LLMs use universal, systematic and interchangeable representations of numbers, we aim to explore whether we can build upon this knowledge in tracing the origins of models' outputs in arithmetic reasoning. First, we assess the predictive power of models' internal representations towards true results in prompts requiring addition, subtraction, multiplication and division.

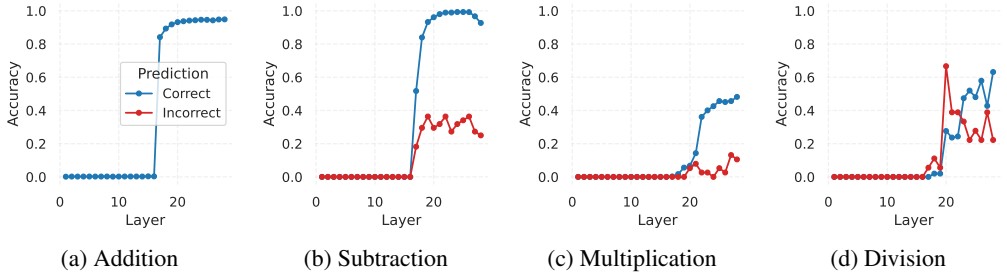

|  (a) Addition | (b) Subtraction | (c) Multiplication | (d) Division |

Figure 9: Probing accuracy in predicting models' outputs for different prompted operations, for cases where the model predicts a correct (blue) and incorrect response (red).

Results of predicting the Llama 3.2 3B's expected outputs for different operations are displayed in Figure 9. We can see a large distinction between the cases where the model predicts a correct and incorrect response – suggesting that the **extent to which the model maintains the sinusoidal**

---

[6]Retrieved from `RealTimeData/bbc_news_alltime`

**representation may determine the accuracy of the output**. We observe that for addition and subtraction, probes can, with close to 100% accuracy, identify the retrieved result already from the model internals. Based on qualitative assessments of output embeddings (subsection B.1), we hypothesize that lower reliability in other operations (multiplication and division) may be caused by the models' divergence from sinusoidal representation employed by the input/output embeddings, accompanied also by lower accuracy of models overall (Table 2).

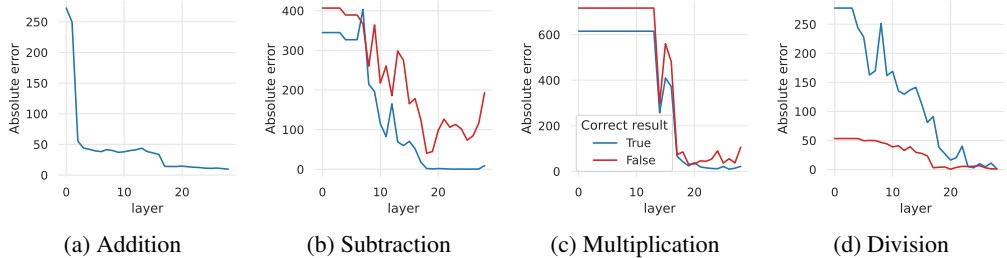

(a) Addition        (b) Subtraction        (c) Multiplication        (d) Division

Figure 10: **Absolute error per layer:** Absolute error of numeric values probed from different layers.

The results in Figure 10 show that across all operations, models tend to incrementally reduce the error towards the true answer value, *gradually* refining across layers. These results also suggest that there are particular layers responsible for an increase of errors in the model's internal computation.

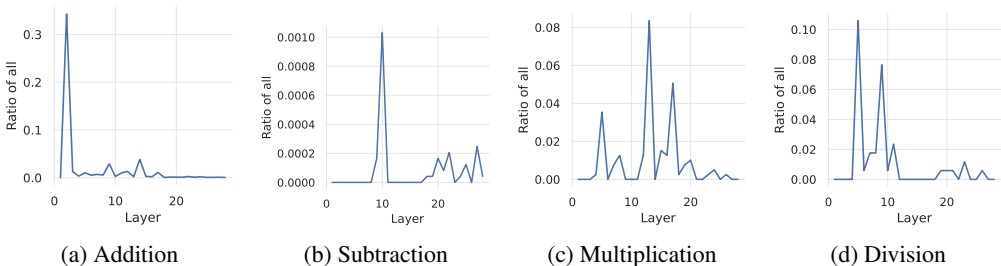

(a) Addition        (b) Subtraction        (c) Multiplication        (d) Division

Figure 11: **Error aggregation per layer:** Relative ratio of cases where each of the layers in Llama 3.2 3B *breaks* the correct result from the previous layer. We find that removing the three layers with the highest error aggregation in division (layers 5, 9 and 11) leads to 27–64% error reduction.

Figure 11 corroborates this hypothesis — showing that particular layers break the correct result probed from the previous layer in large proportions of all predictions. For instance, in division, the 5th layer is responsible for breaking the correct result recoverable from the previous layer in over 10% of all cases. Our further analyses, overviewed in Table 2, show that according to the probed representations, the models often achieve a correct result *internally*, even though it does *not* surface: this is the case of 56.8% of all surfaced (prediction) errors in subtraction, 26.3% in multiplication, and as much as 94.4% of all errors in division.

**Layers ablation** We further support the claimed responsibility of specific layers on the resulting accuracy by simply removing the suspected layers from the model. Such a naive alternation becomes viable with the uncovered universality of representations across layers (subsection 3.3). Specifically, we try to remove one of the three layers with the highest error aggregation, separately in multiplication and in division (having a potential for improvement of accuracy). We find that this brings performance benefit in four out of six cases – in one case (layer 4) in multiplication, causing a 26% error reduction (from 90.38% to 92.91% and in *all* cases in division, with error reduction between 27–64%. However, we must note that the absolute number of correct predictions probed in the lower layers is negligible, which hinders the general applicability of this methodology.

**Steering towards sinusoidality** Taken together, these results underline a hypothesis that the sinusoidal representation is also universal for models' output generation. We can show this more

directly by *steering activations* towards the expected sinusoidal representations. Concretely, we (i) fit sinusoidal probes at each layer to predict the output of a given arithmetic operation; (ii) optimize a set of 1000 randomly-initialized embeddings $\mathbf{e}_y$ to maximize their fit to an optimal representations to a vocabulary of numeric tokens according to these sinusoidal probes; and (iii) steer activations towards the corresponding embedding optimized with respect to the probe whenever the model is not producing the expected output. Steering is achieved by interpolating between the activation $\mathbf{h}_i$ at layer $L_i$ and the optimized embedding $\mathbf{e}_y$ for the intended target scalar result $y$, $\alpha\mathbf{h}_i + (1-\alpha)\mathbf{e}_y$.

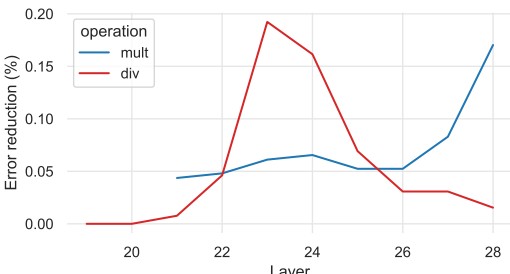

Figure 12: Error reduction after steering activations in incorrectly processed cases of multiplication and division in Llama 3B using interpolation factors of $\alpha = 0.375$ for division, and $\alpha = 0.3$ for multiplication.

Figure 12 displays the absolute difference in performance caused by the steering on previously erroneous cases in multiplication and division. We target probes on upper layers which reach accuracy scores $\geq 10\%$. In short, Figure 12 shows that depending on the layer, we can correct up to $19.23\%$ of errors for division, and up to $17.03\%$ for multiplication.

In summary, factoring in the knowledge of the sinusoidal nature of number representations allowed us to understand and assess the lower bounds of accuracy of multi-token number representations. On a case study of Llama 3.2 3B, we further showed how more accurate probes allow pinpointing and *eliminating* the sources of errors via (i) addressed ablations in the model architecture and (ii) by steering the activations on error cases towards sinusoidality.

## 5 CONCLUSIONS

This paper builds a fundamental understanding of how LLMs internally represent and manipulate numeric values. We provide evidence that diverse LLMs learn and employ mutually-interchangeable, consistent sinusoidal representations across their internal layers, maintained largely by residual streams, but distinct in the embedding representation and internal activations in terms of sparsity. We show how probing techniques respecting representational properties of numbers open up new possibilities for tracking the causes of errors in models' internal computation, attributing large portions of errors to numeric interventions of concrete layers. Towards the fast-growing field of interpretability research, our work contributes by evidencing the importance of natural-language probes in training better-generalizing probes or the superior quality of specialized probes compared to widely-used methods such as linear or TunedLens probes. We hope that our work clearly outlines the potential of more accurate probes in robustness and faithfulness assessments of existing and future reasoning models.

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

## A  SUPPLEMENTARY RESULTS

### A.1  SELECTING THE OPTIMAL NUMBER OF FOURIER COMPONENTS FOR COMPARISON

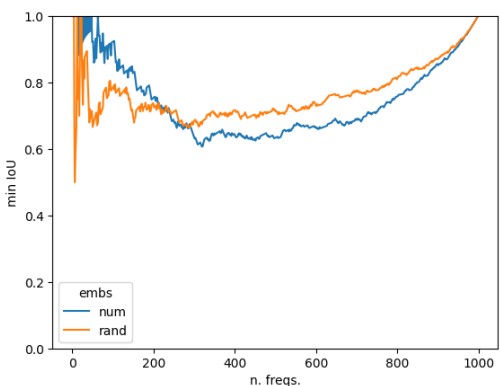

Figure 13: Minimum IoU across number of frequencies considered

In Section 3.1, we discuss selecting the optimal number of Fourier components. As can be assessed from 2a, from a certain frequency rank, Fourier decompositions resort to noise. Thus, our objective is to find the "cutoff" rank ($k$) that disentangles oscillating information from the non-oscillating, presumably non-numeric information. This means finding an *optimal*, yet non-trivial (e.g. $\geq 10$) cutoff for a number of components. In practice, we evaluate for each value of $k$ what the minimum IoU agreement score across all models amounts to for numbers and random overlapping wordpieces. As displayed in Figure 13, we find such an optimal cutoff at $k = 63$ as the highest value of $k$ that leads to perfect agreement across models.

A more thorough assessment of Figure 13 suggests a few interesting trends. Random word pieces also tend to favor a handful of Fourier basis components; which we conjecture is due to the sampling mechanism. By selecting overlapping pieces, our pieces must be frequent enough to be present in multiple distinct tokenizers, which in turns shapes the type of linguistic units represented in this random sample. Secondly, we also observe a 'cross-over' point around $k \approx 250$, after which we find greater agreement in random pieces than numbers. Yet, we note that (i) a significant proportion of the mass is concentrated in a few frequency components for numbers, whereas random pieces lead to much more uniform distributions across frequency components (see Figure 14); and (ii) the results in Figure 13 still allow us to establish that the Fourier profile of number pieces is clearly distinct from what we observe for any other overlapping set of word pieces.



Figure 14: Contributions of Fourier base frequencies for non-number pieces (Llama-3 1B, provided as an example)

## A.2 GENERALIZATION OF PROBES TO UNSEEN LAYERS

Figure 15 shows how probes fitted on each layer generalize to all other layers in a model. Strong cross-layer generalization indicates a high consistency of representations throughout the model.

## A.3 VISUALIZATIONS OF INTERNAL ACTIVATIONS

We visualize the internal activations for a string template "x1 + x2 =" on the second numeric token (x2). Values for x2 are selected as a range 0-999 (all values present by the vocab), and values for x1 are sampled randomly from the same range. Then, we project the activations of the model's middle layer to 64 dimensions with PCA and compute the Fourier transform.

We visualize the first 16 PCA dimensions in Figure 16 and the maximal magnitudes of the frequencies in the Fourier transform in Figure 17. We visualize across model sizes and families.

## B NATURAL LANGUAGE DATASET DESCRIPTION

Table 1: Dataset specifications for numerical embedding analysis across natural language contexts.

| Domain | Dataset | Source | Numerical Context | Size |
|---|---|---|---|---|
| Culinary | Recipe NLG Lite | m3hrdadfi/recipe_nlg_lite | Quantities, measurements | 6118 |
| | FoodRecipe-ImageCaptioning | samsatp/FoodRecipe-ImageCaptioning | Ingredient amounts | 719 |
| Temporal | TimeLineExtraction | irlabamsterdam/TimeLineExtraction...CASE | Legal document dates | 50 |
| Arithmetic | MetaMathQA | meta-math/MetaMathQA | Mathematical reasoning | 395K |
| | DROP | ucinlp/drop | Discrete reasoning | 77.4K |
| | AQuA-RAT | deepmind/aqua_rat | Algebraic word problems | 97.4K |
| Medical | ICD-10 Codes | atta00/icd10-codes | Diagnostic codes | 25.7K |
| | ICD-10-CM | Gokul-waterlabs/ICD-10-CM | Medical classifications | 74K |

| | Add | Sub | Mul | Div |
|---|---|---|---|---|
| Accuracy | 100% | 99.8% | 90.4% | 5.9% |
| P(Extracted \| Incorrect) | - | 56.8% | 26.3% | 94.4% |
| P(Not extracted \| Correct) | 1.4% | 0.03% | 26.1% | 5.9% |

Table 2: **Probe accuracy in extracting the predicted result:** Ratio of cases where the sin probe of some layer (top) retrieves a correct result when the model's prediction is *incorrect*, and (bottom) can *not* retrieve a correct result when the model's prediction is *correct*.

### B.1 QUALITATIVE ASSESSMENT OF OUTPUT EMBEDDINGS

We visualize output representations of Llama 3 1B on addition and multiplication operations. For each expected result value $y$ in the range 0-999, we sample a random pair $(x1, x2)$ from the same range, such that $x1 + x2 = y$ (or $x1 \times x2 = y$, respectively). We then predict the next token for prompts "$x1$ plus $x2$ is " and "$x1$ multiplied by $x2$ is " and collect the final output representations of the model before decoding. We then reduce the representations with PCA to 16 dimensions and visualize the result in Figure 18.

## C EXPERIMENTAL DETAILS

We refer the reader to the companion code-base, which tracks exact hyperparameters for all experiments; the overview provided here is mainly designed for general informative purposes rather than precise replication.

In most experiments, we use an Adam optimizer with a learning rate of $10^{-3}$, and train probes up to 50,000 steps, with an L1 regularization of $10^{-3}$.

In multi-token decoding (Figure 7), we use a learning rate of $5 \cdot 10^{-4}$, 10,000 training steps, and early-stop training.

When fitting probes on natural language contexts (Figure 4, 'Nat. lang. $\rightarrow$ Nat. lang.') and for cross-layer transfer (Figure 15), we use a learning rate of $10^{-4}$.

When dealing with natural language contexts, as well as division and multiplication Section 4.2, we use a learning rate scheduler decreasing the learning rate by a factor of 100 over the first 30,000 steps. Experiments are performed on the probe that maximizes accuracy on a heldout validation set.

In our steering experiments (Figure 12), we optimize embeddings with respect to the probe using an SGD optimizer with a learning rate of $0.1$, optimize the embeddings for 200,000 steps, and decrease the learning rate by a factor of 100 over the first 100,000 steps.

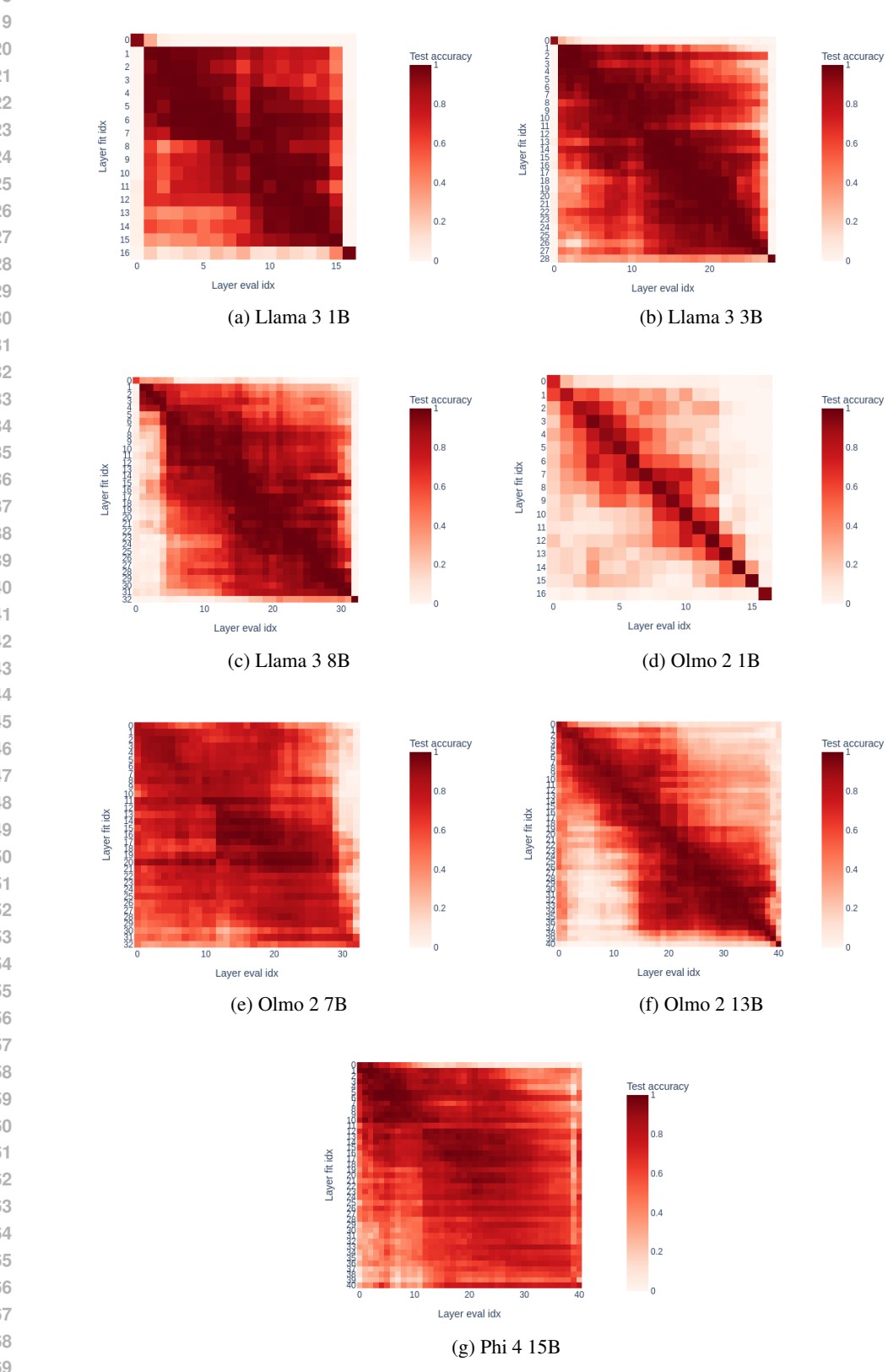

(a) Llama 3 1B

(b) Llama 3 3B

(c) Llama 3 8B

(d) Olmo 2 1B

(e) Olmo 2 7B

(f) Olmo 2 13B

(g) Phi 4 15B

Figure 15: Probe fitted on one layer evaluated on all layers. Olmo 2 1B shows the weakest cross-layer generalization among all models. Llama models display a strong separation between input/output embedding representations and the hidden representations.

(a) Llama 3 1B (layer 8/17)

(b) Olmo 2 7B (layer 16/31)

(c) Phi 4 15B (layer 20/41)

Figure 16: PCA of models' internal representations

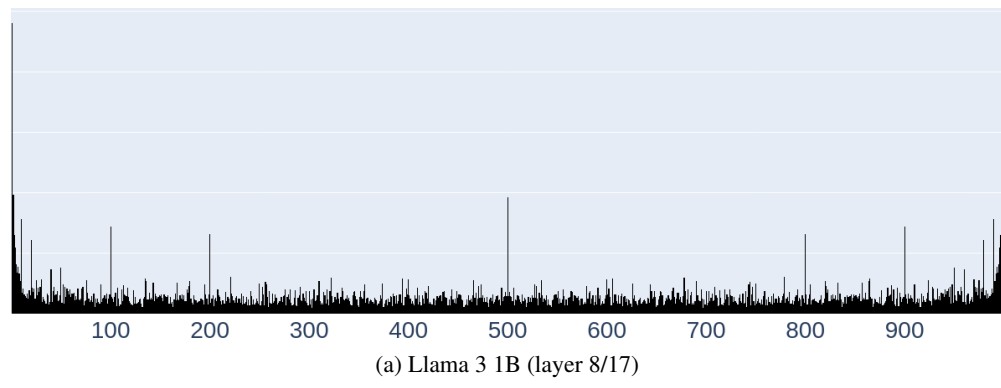

(a) Llama 3 1B (layer 8/17)

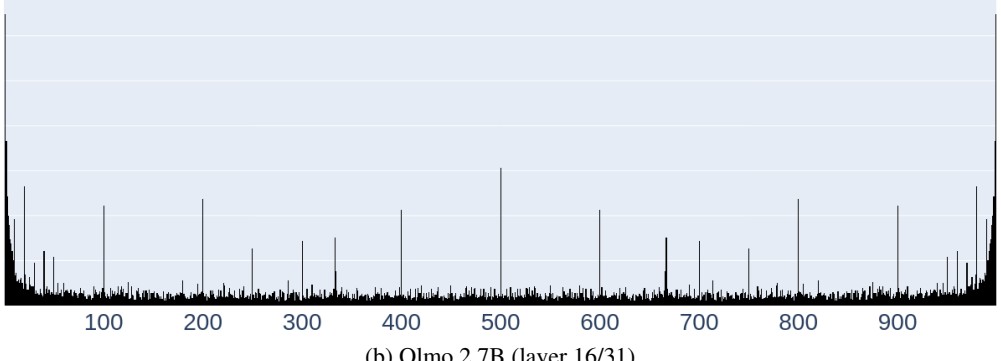

(b) Olmo 2 7B (layer 16/31)

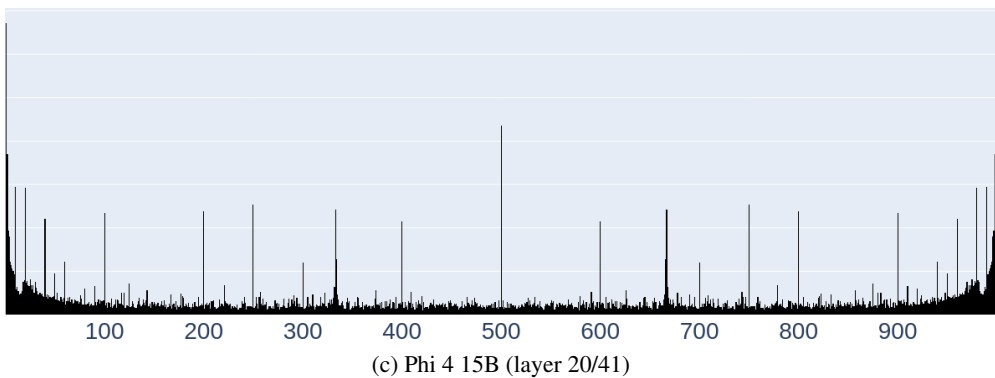

(c) Phi 4 15B (layer 20/41)

Figure 17: Maximal magnitudes of frequencies in Fourier transform of PCA of models' internal representations

(a) addition

(b) multiplication

Figure 18: PCA of Llama 3 1B *output* representations for the operations of addition (recoverable with close-to 100% accuracy) and multiplication (recoverable with appx. 50% of accuracy)

