# OpenReview forum: "Unravelling the Mechanisms of Manipulating Numbers in Language Models"
_ICLR.cc/2026/Conference — ICLR 2026 Conference Withdrawn Submission_

### Official Review · Reviewer_1Ro7 · 2025-10-30

**Soundness:** 2
**Presentation:** 4
**Contribution:** 2
**Rating:** 2
**Confidence:** 3

**Summary:**

This paper presents a variety of observations on how LLMs represent numbers. Namely: that they utilize sinusoidal representations, that representations across nearby layers are similar, that multi-token numbers are represented in the final token, and that accurate responses to arithmetic questions can be localized within the model even when they are not reflected in its final output. To produce these observations, the authors primarily rely on a sinusoidal probe that injects a sinusoidal inductive bias into a linear classifier. They also make some use of TunedLens in the section that considers multi-token numbers.

**Strengths:**

It is an interesting and important question to ask how LMs internally represent numbers and how they use these representations to make next token predictions. The authors present a set of findings that are broadly informative of these questions. Their exploration of multi-token numbers was particularly interesting and a novel contribution, despite my reservations about its interpretation.

**Weaknesses:**

A central goal of this paper is to shed light on how LLMs manipulate numbers internally, in order to understand why they produce erroneous net token predictions in spite of their seemingly accurate number representations. I am not sure that these results have moved the needle on this question for two main reasons:

1. No mechanistic claim about LLMs is complete if it relies only upon correlative evidence, e.g. probing. Since the point of this paper is to make a mechanistic argument rather than to present a probing technique, the lack of causal manipulations significantly detracts from its claims. It is entirely possible that LLMs form representations that can be picked up by a sinusoidal probes even when the underlying mechanism has nothing to do with a sinusoidal representation. Similarly, it is entirely possible that a representation for multi-token numbers can be found within the activations of a single token position even when such a representation has nothing to do with the model's actual manipulation of multi-token numbers. To make arguments about the mechanistic basis of behavior, the authors must be able to demonstrate how mechanism influences behavior.
2. The paper ultimately doesn't say anything substantive about why LLMs make errors despite possessing apparently precise internal representations of numbers. While the broad overview of results is informative, it has not been tied together into a coherent picture of what one should really take away from the paper with respect to this motivating question.

Some other notes:
- I am surprised by the finding that the linear probe in fig 3 has almost 0% accuracy, even at the first layer. That is actually not super believable to me and some additional context there might be helpful.
- Similarly I am confused about the sudden jump in fig 9 from 0% probing accuracy to much higher probing accuracy in a single layer. This is also somewhat suspicious.
- The explanation of figure 7 is confusing -- is the offset the token position in the sequence, or the total token length of the number?
- There is a claim in 4.2 that 'lower reliability in other operations may be caused by the model's divergence from sinusoidal representations'. The authors could investigate this by showing results for other probes. Generally it was not clear why the authors stopped using other probes beyond section 3, since it seems like it would be worthwhile to keep that comparison going.
- The first sentence in section 3.2 'In subsection 3.1 we presented evidence that the input embeddings have certain properties related to how they encode numeric information such as being sinusoidal' seems inaccurate given that there is no mention of sinusoidal representations in section 3.1

**Questions:**

- What should I take away from the notion that representations could be sinusoidal? Why should I care specifically about a sinusoidal probe specifically, versus a more general nonlinear probe?
-

---

> ### Author Response · Authors · 2025-11-28
>
> We thank the reviewer for their thorough feedback!
>
> Regarding Weakness 1.:
> - We now realize that the choice of title might imply too heavily a mechinterp approach, which is not precisely the methodology we are adopting here. Our main point is to show that number representations are systematic,universal and strikingly accurate across models, contexts and layers and provide a series of analyses supporting this. We would appreciate it if the reviewer could point us to specific claims that they believe require mechanistic support in this scope.
> - Striving for open-mindedness, we identified that our claims in Sec. 4.2 could benefit from the mechanistic evidence. Therefore, we added two mechanistic experiments in Sec. 4.2, showcasing (i) what happens if we skip the layers where we evidence aggregate errors in numeric computations, as well as (ii) how to leverage the knowledge of sinusoidality captured by our probes to steer Llama 3.2 3B to mitigate error rates on arithmetic problems. In summary, we show that this can lead to substantial accuracy improvements, particularly in division, exhibiting 27–64% error reduction when skipping one of the three most error-aggregating layers, and 17–19% error reduction with steering.
>
> Regarding Weakness 2.:
> The fact that models do possess precise representations says a lot about where the errors do **_not_** come from — the cause of errors is models’ imprecise **combination** of these representations rather than the representations themselves. This insight may steer and motivate future work towards analysing and mitigating this flaw, while restraining other work from fixing a non-existent problem of accuracy of representations.
>
>
> > I am surprised by the finding that the linear probe in fig 3 has almost 0% accuracy, even at the first layer.
>
> We are evaluating performances in strictly held-out conditions, where the target tokens (i.e. numbers) have not been seen during training. We identify that a fundamental problem of linear probes is their proneness to overfitting – while they reach almost 100% accuracy on the train set, they never achieve improvements over a couple of percent on the validation set. Sinusoidal probe introduces an inductive bias specific to numeric representations that can circumvent overfitting. With infinite data, linear probes would likely achieve better results, but their overfitting is not merely a technical problem – as the qualitative analysis of representations reveals, the linear prior is simply not adequate for modelling numeric-specific representations.
>
> > I am confused about the sudden jump
>
> Remark that Fig. 9 corresponds to probing for the model’s eventual prediction, i.e. the  _output_ of the operation. The plot suggests that there are concrete layers at which point arithmetic computations are being carried out (or have been successfully carried out). See also [Mamidanna et al, (2025)](https://arxiv.org/pdf/2509.09650) for an independent analysis showcasing that arithmetic operations are implemented at concrete, identifiable layers.
>
> > it was not clear why the authors stopped using other probes beyond section 3,
>
> This was an attempt to keep the set of results more focused and less overwhelming, contextualized by vastly inferior accuracies of other probes. In response to this point, we have evaluated the linear and binary probes on previous-token recovery (Sec 4.1) and output recovery (Sec 4.2). However, in both of these cases, neither of these probes is able to provide a relevant insights – the best performance we see are from the binary probes in previous-token recovery, achieving 1.2% of accuracy on -1-th recovered token (c.f. Fig. 7). Therefore, for the interest of sustaining the reader's focus, we have decided not to add these results into a revised version. We newly mention this point in footnote 5 of the current version of the paper.
>
> > The first sentence in section 3.2 'In subsection 3.1 (...) such as being sinusoidal‘
>
> Thanks for spotting this leftover from an earlier version of the draft! We have corrected the sentence in question.
>
> > What should I take away from the notion that representations could be sinusoidal?
>
>
> Please refer to our general response for examples of concrete implications of our work and, we believe, an accurate framing of our contribution. In summary, the knowledge that numeric representations across language models are systematic, predictable, universal and strikingly accurate opens up new opportunities for errors tracking and mitigation (merely pivoted in our Sec 4.2) but also enables future work in interpretability, allowing us to understand to what extent are the internal mechanisms of LLMs robust and reliable.

---

### Official Review · Reviewer_aHXX · 2025-10-31

**Soundness:** 3
**Presentation:** 2
**Contribution:** 2
**Rating:** 4
**Confidence:** 4

**Summary:**

This paper investigates how LLMs internally represent and manipulate numeric information. The authors show that many different LLMs converge to almost identical sinusoidal representations of numbers, not only in their input embeddings but also across hidden layers and in natural-language contexts. Using sinusoidal probes, they study these representations layer by layer and demonstrate that they are highly systematic, transferable across layers, and robust across models. The paper further uses these probes to trace arithmetic computations inside the model and identify specific layers that introduce errors, even when earlier layers have already computed the correct result. The method requires no retraining and adds minimal overhead. Experiments on multiple models and numeric tasks reveal a universal structure in how LLMs encode numbers and show that many arithmetic failures originates from particular layers rather than from poor number representations.

**Strengths:**

1）This work provides several findings about how LLMs represent numbers with extensive experiments across LLMs of different sizes and families.
2）This work applies multiple analysis tools to validate the findings (e.g., RSA, probing, error tracking).

**Weaknesses:**

1）Regarding the unravelled numerical processing mechanism, I'm unclear about its practical implications. Could this understanding be leveraged to systematically improve LLMs' performance on mathematical tasks?
2） It uses extensive experiments to demonstrate the existence of sinusoidal representations, but it‘s limited to explain why this phenomenon occurs. To further explore the inner reason behind phenomenon is essential to better understand LLMs.
3） Do these findings extend to languages such as Chinese and Japanese, where number tokens follow very different segmentation patterns? What could explain the emergence of this phenomenon across languages with distinct tokenization schemes?

**Questions:**

see weakness

---

> ### Author Response · Authors · 2025-11-28
>
> We thank you for the feedback!
>
> As for 1), do take note of our general reply. We have added two sets of experiments to demonstrate that 1) we can leverage the properties we underscore, such as transferability across layers, to improve accuracy; and 2) we can highlight that lower performances on multiplication tasks can be in part imputed to a mismatch in the type of number representation they rely on.
>
> As for 2), we can only offer a tentative conjecture as to why this phenomenon arises.
> We can break down our observations into three major points: 1. the type of representation (sinusoidal), 2. their universality across models, and 3. their systematicity across layers.
> 1. As we briefly highlight in the state of the art, [Nanda et al. (2023)](https://openreview.net/forum?id=9XFSbDPmdW) provide a solid explanation of why modular mathematics can be represented using trigonometric representations, which is relevant when producing the next number token (in a 0-999 integer range).
> 2. As for universality across models, we should note that all models in this study are text-based, English-centric LLMs trained on corpora that often overlap to a non-trivial extent. We can also expect that the distributional properties of numbers are fairly well-delineated, and that therefore we should end up with comparable representations. After all, simpler static embeddings for different languages can be aligned based on a few word translations (e.g., [Lample et al., 2018](https://openreview.net/forum?id=H196sainb)), and therefore it's viable that a relatively homogeneous set of models yields representations that are fairly similar for a well-delineated class of tokens.
> 3. Lastly, the fact that representations are very stable across models might be due to the presence of a residual stream, as pointed out by reviewer uzDK.
>
> There are counter-arguments to all three of these points, which is why this is at best a very tentative conjecture as to how this phenomenon arises.
>
> As for 3): indeed, we do not expect the pattern to emerge in Chinese or Japanese. Broadly speaking, we expect the situation to be stable across uses of Arabic numerals in all languages, as long as the tokenizer is set up to have individual tokens for all integers in a stable range and there is sufficient training data for the pattern to emerge. For the specific situation of Chinese and Japanese, this set of constraints might not be met.

---

### Official Review · Reviewer_uzDK · 2025-11-01

**Soundness:** 2
**Presentation:** 2
**Contribution:** 2
**Rating:** 2
**Confidence:** 4

**Summary:**

This paper extends a body of prior work showing that LLMs use sinusoidal/periodic features to represent numbers and perform arithmetic operations. They find a very high agreement between the representations learned by different models to represent numbers.

**Strengths:**

The results on generalization of probes are interesting and confirm that sinusoidal probes are more effective than other natural alternatives.

Some intriguing results are presented, for example comparing probe accuracy across layers and investigating how information about multi-token numbers can be extracted.

**Weaknesses:**

Overall the paper seems quite incremental. It is already known (as the paper states) that different LLMs learn the same type of number representations. This paper goes further by measuring exactly how closely the representations match, but this seems like only a small step in terms of contributing new knowledge. The findings on probe generalization, while interesting, recapitulate known findings about probes in general (e.g., https://arxiv.org/abs/2410.02707 https://arxiv.org/abs/2506.00823).

Other findings are somewhat intriguing but it's not clear what the significance or takeaway is. For instance, Figure 6 (difference in probe weights) is interesting, but it is not clear what the significance of this finding is. The same goes for the multi-token and error tracking experiments--it's not clear what is gained by knowing, e.g., which types of layers contain information about previous tokens. For error tracking, ideally there would be a way to anticipate whether the model will make wrong predictions, or explain why it makes wrong predictions, but the current results fall short of this.

The consistency across layers (e.g., Line 353) is not very surprising given that the residual stream maintains a running sum.

**Questions:**

I noticed that there are Zhou et al. 2024a and 2024b, but they're both for the same paper.

Line 078: I don't think it's correct to say that Zhou et al. "recast" the observations of Kantamneni and Tegmark, since the Zhou paper appeared first. It's more correct to say that Kantamneni and Tegmark recast the observations of Zhou et al.

Line 388: I was confused by why you used the 2000-3000th most common tokens, and not the top 1000 tokens. Alternatively, why not just choose all the number tokens in the vocabulary?

Line 392: Instead of probing pieces of multi-token numbers, could you probe for the overall value of the multi-token number?

For error prediction, you are measuring whether the probe matches the model's final outputs. What if you instead plotted the probe's accuracy relative to the correct answer? Is it possible that for the red lines, the probe is predicting the correct answer but the model is incorrect?

---

> ### Author Response · Authors · 2025-11-28
>
> Thank you for your time giving us a review! Please take a moment to consider our responses below.
>
> > It is already known that different LLMs learn the same type of number representations.
>
> The case made in this paper is much stronger than what prior literature, including the referred one, shows — prior work shows that numbers tend to be represented in a wave-like pattern. We show that the exact same wave-like arrangement (up to minor transformations that we analyze and describe) holds **throughout** the model and, crucially, remains highly accurate (note that what we are showing are the lower bounds of this accuracy).
> We also show that the representations of numbers for different models are semantically interchangeable – i.e. facilitate a decomposition into identical frequencies — evidencing the case of the Platonic representation hypothesis ([Huh et al. (2024)](https://proceedings.mlr.press/v235/huh24a.html)) with numbers.
>
> > The consistency across layers (e.g., Line 353) is not very surprising given that the residual stream maintains a running sum.
>
> The residual steam can, but *does not need to*, maintain this consistency and previous work indeed shows cases disputing this assumption: the individual terms in the running sum of the stream can and do vanish due to the reweighting during layer norms; e.g. [Mickus et al. (2022)](https://aclanthology.org/2022.tacl-1.57/) show the input embeddings have a minimal contribution to the running sum after the first few layers in models as simple as BERT. Therefore, the wave-like properties could also be discarded, if e.g. any given sublayer adds a non-wavelike term to the running sum with a sufficiently large norm.
>
> > Other findings are somewhat intriguing but it's not clear what the significance or takeaway is.
>
> Our main takeaway is that the **same, well-predictable representation arises across different LLMs, different layers and different contexts** — whenever LMs manipulate numbers. Please see also our general response for implications and, we believe, accurate framing of our contribution.
>
> > What if you instead plotted the probe's accuracy relative to the correct answer? Is it possible that for the red lines, the probe is predicting the correct answer but the model is incorrect?
>
> This choice is motivated by the assumption that the numeric information conditioning the real output must be present in the model computation, and therefore, must be recoverable from some hidden state. This is not the case for the ground-truth answer, which, in the case of incorrect answers, may never be present. As you correctly assume, in this setting, the probe may predict the correct answer even if the model is incorrect — and this is, as we find, indeed the case for the large portion of model's errors; in division, as much as 94.4% (Appx B, Table 2).
>
>
> ### Questions
>
> > Zhou et al. 2024a and 2024b
>
> Thank you for the comments! We have corrected the duplicate reference.
>
> > why you used the 2000-3000th most common tokens
>
> Our aim here was to make the comparison with natural-language tokens’ probing as fair as possible. The distribution of numbers in numbers’ probing setting was uniform, and therefore, we tried to achieve a similar distribution also with natural-language tokens, but still focusing on language tokens with a common occurrence in language. As these tokens follow Zipfian distribution, we refrained from using the head of the distribution and thus, used the 2000-3000th most common tokens, that are already distributed relatively uniformly — making the probing task similarly difficult to the numeric setting and thus, well-comparable.
>
> > could you probe for the overall value of the multi-token number
>
> The reason why we chose our current setting is that the target number of classes (1000) remains tractable and feasible to train for, as opposed to using the exact value of three-token numbers ($10^9$ classes). We're open to refinements, but we believe that in the current state, our analysis has the same power as the per-token predictions can be bijectively mapped onto even any multi-token numeric value.

---

### Official Review · Reviewer_gZab · 2025-11-02

**Soundness:** 3
**Presentation:** 2
**Contribution:** 1
**Rating:** 2
**Confidence:** 3

**Summary:**

This paper presents a series of experiments about properties of the embeddings and internal activations of numbers in large language models. Here is an overview of the key results:

1. The paper shows that different LLMs learn input embeddings for numbers that have similar relative structures, as measured by a Representational Similarity Analysis.

2. The paper shows that the PCA’d coordinates of number embeddings for different LLMs have similar highly weighted fourier frequencies.

3. The paper shows that across different models, linear probes trained with a sinusoidal prior perform much better than linear probes trained with other priors at the task of predicting the number associated with network activations at all parts of the residual stream.

4. The aforementioned sinusoidal probes are shown to be able to predict numbers in both natural language and numerical settings.

5. The paper shows that an aforementioned sinusoidal probe trained on one layer can generalize to nearby layers.

6. The paper shows that the strongest performance of sinusoidal probes comes from the residual stream, rather than intermediate attention and MLP outputs.

7. The paper shows that sinusoidal probes can also be used to predict earlier number tokens from the last-token activations of a multi-token number. However, the accuracy of the probes decays as you try to predict earlier and earlier number tokens.

8. The paper shows that the aforementioned sinusoidal probes can be used to predict the answer to simple arithmetic questions, and that these probes perform better when the model gets the answer correct vs. when it gets the answer wrong, and that the probes generally have better performance at later layers in the network.

**Strengths:**

S1: The paper includes many different experiments providing insight into various aspects of number embeddings and activations in large language models.

S2: The RSA experiment was a very elegant way of showing how different models have similar embedding structures.

S3: The paper very clearly references prior work on top of which it builds on.

**Weaknesses:**

W1: The biggest weakness is that the paper contained a lot of thematically related results, but no concrete cohesive framework to link the results together. In particular, it’s unclear if these results enable a practitioner or theorist to do anything they were previously not able to do.

W2: I found the experiments a little difficult to parse and I had to infer a lot of the experimental details. I think spending more time being precise about the exact setup for the experiments would help.

**Questions:**

Questions

* Q1: For the sinusoidal probe experiments, which token indices did you train and test the probe on?

* Q2: For the multi-token numbers experiment in section 4.1, did you train new sinusoidal probes for this task or use the ones described in previous sections?

* Q3: For the linear probing experiments, the sinusoidal probe is just a special type of linear probe with a particular prior. I would expect that in the limit of infinite training data, a regular linear probe should also be able to match the performance of the sinusoidal probe. Is this true, or am I misunderstanding something? If I am not misunderstanding something, I would be curious to see some scaling laws or rough estimates of how much training data is necessary for the linear probe to come close to the performance of the sinusoidal probe.

Suggestions

* SU1: To address weakness W1, the authors could comment on or demonstrate concrete applications of their observations. The most compelling applications are ones that are not previously achievable without the insights gained from this work.

* SU2: It would be helpful if you clarified the dimensions of the matrices in the sinusoidal probe in lines 174-178. I eventually figured out the dimensions by consulting prior work, but it would have saved time to include them in the paper.

* SU3: I found it surprising in Figure 2(c) that random tokens would have high fourier frequency overlap. It would be interesting to show a version of plot 2(a) also for random tokens.

---

> ### Author Response · Authors · 2025-11-28
>
> We thank the reviewer for their time with our paper and their feedback. Please find our responses, together with a summary of changes, below.
>
> W1/SU1: Please refer also to our general response. In summary, the key point we make in this paper is that number representations are systematic, universal and strikingly accurate (Sec 3). This understanding opens up a myriad of opportunities in early error detection, circuit discovery, or targeted architecture refinements -- of which, we explore some in Section 4.
>
> As a response to your point, we further extend Section 4 by showing how the knowledge of representation universality and error attribution can be useful in errors mitigation – we show that by ablating layers with the largest error aggregation ratio (Fig 11), the model will not only remain functional, but we can mitigate up to 64% output errors in prompts involving division; steering allows us to reduce errors by up to 19%.
>
> As for W2, we accompany all results presented in the paper with fully reproducible Jupyter notebooks, intentionally fixing all the hyperparameters involved. However, in the current revision, we also include all relevant details in the newly-added appendix C and remain open to specifying any further, concrete details in the text as well.
>
>
>
> Q1: We randomly reserve 5% of numeric tokens, i.e. target values for testing. The corresponding values are not seen during training. These values are sampled randomly from the full range of <0; 999> represented as separate tokens.
>
> Q2: We trained a separate probe for this task. Using the same probe would result in recovering the current-token value, which is not our intention — indeed, we find that previous tokens reside in a specific, yet consistent topology in the shared representation space..
>
> Q3: While the classifier probe uses a linear architecture, the sinusoidal bias is applied to all class vectors, including those we hold out for evaluation. In this strict setting of “unseen labels”, a standard linear probe ends up with randomly initialized vectors for the held-out classes, regardless of the number of samples we consider. Also note that we have to respect an effective ceiling to the number of samples in this task (at most 1000 x 1000 for single-token experiments with 1000 labels), which, in our methodology, rules out approaches reliant on raw scaling.
>
> SU2: Thank you for your suggestion. The details are now provided after Eq. 1.
>
> SU3: Please see the added plot (Fig. 13) in Appendix A. When compared to numeric tokens (Fig. 2a),  we can see that while the random language tokens also exhibit higher significance of some frequencies, this trend is much less significant than with numeric tokens.

---

### Author Response · Authors · 2025-11-28
**General response**

Foremost, we'd like to thank all our reviewers for their time dedicated to our work!

We notice that a primary, shared concern among reviewers pertains to the applicability of our work.
First, we'd like to emphasise that our contribution is in *understanding* the properties of representations (Sec. 3). As such, our work is *not* aimed to be directly applied but it opens up very concrete possibilities. In Sec 4, we pivot the exploration of some of these opportunities, but there are many more that our work enables, e.g.:
- Interpreting the internal computation of the models, including the sources of errors — advising future work on "what needs to be fixed" to improve models' accuracy
- Enabling mechanistic alterations of models' decision making, allowing us to obtain control over the models' responses
- Improving the precision of circuit tracing methods
- Assessing the robustness of numeric operations within the LM

We believe that properly establishing (i) that there is a strong prior for any number representation to be sinusoidal; and (ii) that the exact properties of these representations are highly predictable given the numeric value it represents; warrants its own solid, secure foundation and constitutes a well-delineated unit of work. This is what we’re hoping to do with this paper.

Nevertheless, in the interest of supporting the reviewers’ confidence in potential applications of this work, we have employed our findings in two new experiments:
1. We build upon findings in §3.3 and extend our results in §4.2 by mechanistically demonstrating that we can remove specific model layers and not only maintain the functional model, but even reduce the error rate of the model by up to 64%;
2. We show that by steering activations towards the universal sinusoidal pattern that we uncovered, we can mitigate 17–19% of errors on multiplication and division;

Additionally, we have implemented a series of textual refinements that we believe will articulate our contribution more precisely.

---

### Note · Authors · 2026-01-06

**Comment:**

We sincerely thank our reviewers for providing us with expert feedback on our work. We will consider how to most efficiently address your raised points, despite not receiving acknowledgements for the arguments from our rebuttal.

Nevertheless, with the ICLR's decision to ban the discussion, we are sorry to see that our justifications about the framing of our contributions could not have been considered, wasting both our work and that of our reviewers. We do not foresee how this dominant point could be addressed without further discussion, and therefore, we have decided to withdraw.

Our sincere hope is that a similar submission experience — compromising the quality of the reviewing process based on the assumption of guilt — will not be exemplary for any other conferences.

**Withdrawal Confirmation:**

I have read and agree with the venue's withdrawal policy on behalf of myself and my co-authors.